# Development and Validation of a Self-Efficacy Scale for Nursing Educators’ Role in Sri Lanka

**DOI:** 10.3390/ijerph18157773

**Published:** 2021-07-22

**Authors:** Shyamamala S. Weerasekara, Jina Oh, Haeryun Cho, Mihae Im

**Affiliations:** 1Post Basic College of Nursing, Colombo 01000, Sri Lanka; shyamwsm@gmail.com; 2Institute of Health Science, College of Nursing, Inje University, Busan 47392, Korea; 3Department of Nursing, Wonkwang University, Iksan 54538, Korea; chr@wku.ac.kr; 4Department of Nursing, Choonhae College of Health Sciences, Ulsan 44965, Korea; mihae1219@gmail.com

**Keywords:** education, nursing, faculty, teaching, Sri Lanka

## Abstract

This study develops a scale that assesses the self-efficacy of Sri Lankan nursing educators in assuming the roles of nursing educators and validates its psychometric properties. This methodological research followed the DeVellis Scale Development Model, which involves six steps of instrument development and evaluation. Preliminary items were determined through a literature review and focus group interviews with nine Sri Lankan nursing experts. The experts, comprising five South Korean and two Sri Lankan nursing professors, tested the scale’s content validity. Moreover, 15 nursing educators participated in a pilot study, and 126 educators took part in the main survey. To evaluate the scale’s validity and reliability, the data from a preliminary questionnaire were analyzed using SPSS/IBM and AMOS 24.0. Further, construct validity was tested using exploratory and confirmatory factor analyses, and reliability was tested by calculating Cronbach’s alpha and performing split-half testing. Finally, 39 items under four themes, “clinical mentorship” (18 items), “research” (10), “teaching” (6), and “advising” (5), explained 63.5% of the total variance. Confirmatory factor analysis results revealed an acceptable model fit for the final scale. The developed scale achieved a Cronbach’s alpha value of 0.97. Thus, the psychometrical properties of the scale measuring Sri Lankan nursing educators’ self-efficacy were comprehensively evaluated and found acceptable. The developed scale will be useful in guideline development or studies regarding the self-efficacy of nursing educators’ roles in developing countries with similar context to Sri Lanka.

## 1. Introduction

Faculty qualification is one of the major considerations in nursing education [1], and provision of support to enable novice educators to become capable of playing higher roles in nursing education is very important [2]. The nursing school program in Sri Lanka serves as a transitional stage between the three-year diploma and four-year bachelor’s degree courses [3]. The country has basic nursing schools that offer a diploma-level nursing program; among them, 19 national basic nursing schools are called nursing training schools (NTSs). The nursing program offered by NTSs provides practice-oriented education and involves more than 7000 practicum hours. The country’s Post Basic College of Nursing for registered nurses (RNs) is the national post-qualification nursing educational institution offering courses equivalent to the RN bachelor of science in nursing (RN-BSN) program [4]. These nursing programs are offered not only in Sri Lanka but also in developing countries such as the Maldives and Laos. The nursing educators who teach in nursing schools offering diploma-level programs face many challenges in their work role transition to nursing educators [3,5].

The Southern Regional Education Board (SERB) identified the following three roles of nursing educators: teacher, scholar, and collaborator [6]. Moreover, according to Meretoja et al. [7], nursing educators’ competencies include functional adequacy and the capacity to integrate nursing students’ knowledge, skills, attitudes, and values in specific contextual situations in both the nursing classroom and real-world clinical settings. Sri Lankan healthcare authorities should more consciously develop the competencies of allied healthcare professionals, including nurses, today more than ever before to overcome the challenges posed by advanced healthcare environments [7]. Nursing educators play a key role in education by working with clinical nurses to prepare and support the educating for nursing students [2]. Hence, this study focuses on nursing educators’ self-efficacy from the perspectives of nursing educators in Sri Lanka.

The educators must have self-efficacy, that is, the belief that their teaching performance significantly influences their students’ academic achievements [8]. A nursing educator’s self-efficacy is the instructor’s belief in the ability of nursing students to implement and organize the behaviors that characterize competent nurses [1,9]. The roles required to be performed by nursing educators differ from one country to another, and the educators’ academic quality varies among countries [7]. Fukada [10] found that each country has different requirements regarding nursing educators’ roles depending on diverse education systems. Therefore, a questionnaire on nursing educators’ roles appropriate to the conditions of this country was needed [2]. The measurement is a fundamental activity concerned with the measuring of phenomena [11]. Since it is essential to identify the current status of nursing educators’ roles in Sri Lanka, an appropriate measurement scale should be developed to assess the roles of the country’s nursing educators.

In Asian countries, many researchers have developed instruments to assess nursing educators’ roles or competencies, such as the clinical nursing faculty competency or self-efficacy inventory [9,12], clinical nursing competency scale for clinical preceptors [13], academic nursing educator competency [1], and patient safety teaching competency of nursing faculty [14]. However, these instruments cannot be used to evaluate clinical and academic competency in Sri Lanka, whose nursing education system and context are different from those of developed countries. Therefore, the roles of nursing educators suited to the unique context of Sri Lanka require assessment.

The purpose of this study is to identify Sri Lankan nursing educators’ role as nursing educators and develop and evaluate the psychometric properties of a scale that assesses the self-efficacy of nursing educators in performing their roles. The specific research objectives are as follows: (a) develop an instrument to evaluate the Self-Efficacy Scale for Nursing Educator’s Role in Sri Lanka (SSNER-SL) and (b) validate this instrument.

## 2. Methods

### 2.1. Study Design

In this methodological study, the SSNER-SL was developed following the DeVellis Scale Development process [11] and validated (Figure 1) step by step. In this study, each preliminary item was constructed with concept analysis and review pre-existing scales. For ensuring validity, exploratory factor analysis (EFA), confirmative factor analysis (CFA), content validity, convergent validity, discriminant validity, and a scale’s internal consistency, split-half reliability were calculated for ensuring reliability.

### 2.2. Concept Analysis of the Nursing Educator’s Role

Concept analysis was conducted through a literature review and a focus group interview (FGI) at the initial developmental phase of the questionnaire. Initially, the relevant literature on the scales of the nursing educator’s role was reviewed to generate an item pool. The literature review was conducted on the papers published in PubMed and Cumulative Index to Nursing and Allied Health Literature using the keywords “nurs *,” “educa *,” “role,” and “scale”. The inclusion criteria were as follows: the studies had to be published during the past 10 years, in English, and peer-reviewed. All studies targeting clinical practitioners were excluded. The final search date was 30 January 2016. Finally, eight articles and two scales [12,15] related to the nursing educator’s role were selected for concept analysis. The interview’s scope was determined according to the literature review.

Subsequently, the qualitative data collected from Sri Lankan nursing experts working as principals in national nursing schools and from nursing educators were analyzed. The main interview questions were “what do you think about nursing educators’ role in Sri Lanka” and “what do you think about the role required for nursing educators in Sri Lanka?”. In the interview conducted on 20 February 2016 at the Sri Lanka Nursing Council, nine nursing academic experts participated. An in-depth, 90 min interview conducted in English was recorded.

### 2.3. Composition of Preliminary Items

Based on the significant sentences obtained from the literature review and FGI, preliminary questions were obtained. The preliminary questions were formulated using the significant phrases of literature review and the FGI. A clear and simple sentence was used, and each question was written to express a single concept.

### 2.4. Content Validity and Pilot Testing

Content validity testing was conducted from 1 May to 14 May 2016. Seven experts from South Korea and Sri Lanka commented on the scale’s content validation. Apart from being nursing professors and having experience in tool development, the five Korean experts understood the situation of nursing in Sri Lanka, because they have been conducting the Official Development Assistance (ODA) Korea Project for Sri Lanka since 2013 for adopting a specialized and systematic nursing education system in Sri Lanka. Two Sri Lankan experts were nursing educators in the NTS and had more than 10 years of academic experience. The experts reviewed the relevance of each question, and the response scale ranged from 1, not at all, to 4, extremely appropriate. Further, the experts conducted content validity testing using the item-content validity index (I-CVI), scale-content validity index/average (S-CVI/Ave), and S-CVI/universal agreement (S-CVI/UA).

The pilot test was conducted on 15 nursing educators in Sri Lanka on 31 May 2016. Experts reviewed the relevance of each item’s contents to the purpose of the scale and responded using a 4-point Likert scale. Thus, the I-CVI, S-CVI/Ave, and S-CVI/UA scores were obtained for this phase.

### 2.5. Validity and Reliability Testing

#### 2.5.1. Sample and Setting

More than 150 samples were deemed appropriate for each exploratory factor analysis (EFA) [16] and confirmatory factor analysis (CFA) [17]. The study samples represented the entire nursing educator population of 19 national nursing schools in Sri Lanka. Accordingly, 173 questionnaires were distributed, excluding a pilot sample of 15 nursing educators, out of which 133 responded (76.9%). The low response rate was due to the problematic postal system in Sri Lankan. Finally, after excluding 7 incomplete responses, 126 responses were analyzed. Attempts to collect additional data were made by contacting the subjects who provided incomplete responses and those who did not respond at all; however, additional data could not be collected because of an absence of responses. Therefore, EFA and CFA were conducted on the same subjects.

#### 2.5.2. Data Collection

The data for construct validity and reliability testing were collected from 20 July to 20 August 2016. The questionnaire was in English because all nursing educators are familiar with English terminology and, currently, teaching is conducted in English. The researchers maintained frequent communication with the participants both directly and indirectly to collect relevant data. The questionnaires were distributed and collected through mail and by hand.

#### 2.5.3. Validity Testing

Before conducting the EFA, each item’s mean and standard deviation, skewness, and kurtosis were checked to determine the normality of data. Subsequently, corrected item total correlation of each item was checked to confirm its communality with other items and redundancy.

Further, EFA was conducted to confirm the scale’s construct validity. The EFA included principal component analysis and varimax rotation, and the factor structure was assessed using Kaiser normalization [18]. Subsequently, CFA was conducted to assess the latent construct in EFA. The model’s goodness of fit, convergent validity, and discriminant validity were analyzed using CFA, as well.

#### 2.5.4. Reliability Testing

The Cronbach’s alpha value was calculated to measure the internal consistency of the scale. Further, the scale’s internal consistency and split-half reliability were analyzed to test its reliability. The Cronbach’s alpha value was calculated for each factor, and split-half reliability with the first-second half method was used.

### 2.6. Data Analysis

#### 2.6.1. Qualitative Data

To ensure the trustworthiness of FGI data, all the completed and transcribed interviews were returned to the nine participants. Each participant was asked to verify the accuracy of the transcripts. Such participant checks were used to verify the accuracy of the recorded words and the conceptions expressed by the participants and to identify any misunderstandings.

The literature review and FGI data were examined using an analytical framework that was designed to identify the contents and components of nursing educators’ roles. The qualitative data were analyzed using thematic analysis with codebook [19]. The researchers independently examined the data, formulated initial codes and categories, and then compared their findings. The completed codebook included codes, categories, themes, and exemplary narratives. The categories obtained from FGIs were compared with the data obtained from the literature review. In this manner, the nursing educator’s common roles were determined.

#### 2.6.2. Quantitative Data

The quantitative data collected by the questionnaire survey were analyzed to test content validity, construct validity, including EFA, CFA, and convergent validity, and reliability using IBM/SPSS Statistics for Windows, version 24.0 (IBM Corp., Armonk, NY, USA) and AMOS 24.0 (Chicago, IL, USA).

To calculate the scale’s content validity, the number of experts who scored 3 or 4 points was divided by the total number of experts obtaining an I-CVI, and an item with a content validity score of 0.80 or higher was considered a suitable item [20]. S-CVI/Ave was obtained by calculating the average I-CVI value of each item, whereas S-CVI/UA was obtained by estimating the ratio of items evaluated as 1.0 of the total I-CVI value. S-CVI/Ave was determined to calculate the content validity if it was more than 0.90, and the S-CVI/UA value was confirmed for content validity if it was more than 0.80 [21].

Items were analyzed using the mean, standard deviation, skewness, kurtosis, item-total correlation (ITC), and Cronbach’s alpha values after item removal. Further, the items having absolute values of 2.0 or more for skewness and kurtosis were considered abandoned. ITC values more than 0.30 were considered acceptable [22]. The general characteristics and measurement results of subjects were used with descriptive statistics. The collected data were tested for the appropriation of EFA using the Kaiser–Meyer–Olkin (KMO) test, Bartlett’s test of sphericity, and measures of sampling adequacy (MSA) value in anti-image matrices. The factors were determined using the following criteria: they should have a factor loading of 0.40 or higher and an accumulative variance of 60.0% or higher [23].

The goodness of fit of the model for CFA was analyzed using the following measures: chi-square statistics (χ^2^), standardized chi-square statistics (χ^2^/df), the root mean square residual (RMR), the root mean square error of approximation (RMSEA), the goodness-of-fit index (GFI), the normed fit index (NFI), the Tucker–Lewis index (TLI), and the comparative fit index (CFI). The reference values for the fitness indexes were *p* > 0.05 for χ^2^, χ^2^/df ≤ 3.0, RMR ≤ 0.05, RMSEA ≤ 0.10, GFI ≥ 0.90, NFI ≥ 0.80, TLI ≥ 0.80, and CFI ≥ 0.80 [23,24].

Convergent validity was tested using the conditions that the construct reliability (CR) was greater than 0.70 and average variance extracted (AVE) was greater than 0.05. Further, discriminant validity was tested under the condition that the square root of AVE of each factor was greater than Pearson’s correlation coefficient [23].

In addition, reliability was tested using Cronbach’s alpha for internal consistency and split-half reliability of the data. The reliability of the complete scale and that of each factor was considered acceptable if the Cronbach’s alpha value was greater than 0.65 [11].

### 2.7. Ethical Consideration

This study was reviewed and approved by the I University Institutional Review Board (IRB No. 2-1041024-AB-N-01-20160118-HR-342). Participants voluntarily contributed to the study once they were assured of the confidentiality and anonymity of their data and made aware of the study’s nature and purpose.

## 3. Results

### 3.1. Concept Analysis of Nursing Educators’ Role

In the first step of the qualitative analysis of the literature review and FGI data, 84 descriptive statements were extracted from the transcripts. Subsequently, 22 contents statements, 13 categories, and 4 themes were derived. Further, the four main roles of nursing educators, teaching, research, clinical mentorship, and advising, were suggested (Table 1).

### 3.2. Categorization of Preliminary Items

Each statement was reformulated as a question, and each question’s wording and relevance were assessed with minor adjustments. Then, 64 preliminary items of the SSNER-SL were extracted and categorized into the following roles of nursing educators: teaching (22 items), research (15), advising (10), and clinical mentorship (17).

### 3.3. Content Validity and Pilot Testing

The I-CVI of 64 preliminary items varied from 0.50 to 1.0, and the S-CVI/Ave and S-CVI/UA were 0.88 and 0.47, respectively. Based on the content validity results and experts’ recommendations, 13 items were deleted and 8 were revised to improve clarity. Finally, 51 items were selected for the pilot test.

In the pilot test, I-CVI ranged from 0.60 to 1.0, and S-CVI/Ave and S-CVI/UA were 0.94 and 0.75, respectively. The items noted as being vague in terms of understanding and being inappropriate to the Sri Lankan context were revised. The three items were removed since they scored low values of I-CVI: (a) understand legal and ethical issues—related policies, procedures, and guidelines in education; (b) contribute to social services along with healthcare team; (c) maintain the confidentiality and security of professional information. Finally, 48 items with I-CVI ranging from 0.80 to 1.0, an S-CVI/Ave of 0.96, and an S-CVI/UA of 0.81 were selected.

### 3.4. Validity and Reliability Testing

#### 3.4.1. Identification of Participants’ General Characteristics

The data on participants’ clinical and teaching experiences and continuous education, in addition to their basic demographic information, were collected (Table 2). The participants’ mean age was 46.36 years. All the participants had teaching diplomas; however, 28.6% did not have the bachelor’s degrees. The respondents’ average teaching and clinical experiences were 10.50 and 9.87 years, respectively.

#### 3.4.2. Item Analysis

For each item, the mean score was 2.36–3.64, with a standard deviation of 0.76–1.06. The skewness was −0.20–0.43, and kurtosis was −0.92–0.19, which demonstrates the dataset’s normality. Finally, the ITC was 0.43–0.76, and Cronbach’s alpha value after item removal was 0.97.

#### 3.4.3. Exploratory Factor Analysis

To determine the number of the extracted factor, scree plot, parallel analysis, and Eigen value were identified. The scree plot graph showed that the slope plateaued after the fourth point, and parallel analysis was found to be appropriate to be divided into a second factor. There were six factors for which the Eigen value was more than 1. When developing the scale, the theoretical framework should also be taken into accounts, with the statistical criterion [11,18]. In the first stage of scale development, four attributes were identified by concept analysis. Accordingly, the EFA was implemented by designating four factors.

In the EFA’s first phase, the KMO measure of sampling adequacy for the items yielded an index of 0.91, the Bartlett’s test of sphericity provided χ^2^ = 5145.548 (*p* < 0.001), and MSA ranged from 0.85 to 0.95. This indicates that the data were appropriate for EFA. The EFA was conducted using the principal component method with varimax orthogonal rotation and revealed that four factors accounted for 60.96% of the overall variance. The maximum factor loading was under 0.40, or the items were not classified as factors [18]. Therefore, 9 items were deleted, and 39 items were categorized into the four factors that remained after the second EFA.

In the EFA’s second phase, the KMO value was 0.92, Bartlett’s test of sphericity revealed that χ^2^ = 4025.64 (*p* < 0.001), and MSA ranged from 0.86 to 0.96. The communality of all the 19 items was higher than 0.4 and ranged from 0.40 to 0.77. The factor loading ranged from 0.50 to 0.86. The first factor with 18 items, second factor with 10 items, third factor with 6 items, and fourth factor with 5 items explained 27.31%, 16.55%, 10.12%, and 9.60% of the variance, respectively. Finally, the total explained variance was 63.58% (Table 3). Each factor was provided one of the following labels: “Clinical mentorship (factor 1),” “Research (factor 2),”, “Advising (factor 3)”, and “Teaching (factor 4)” (Table 3).

#### 3.4.4. Confirmatory Factor Analysis

The multivariate skewness was −0.35–0.43, and multivariate kurtosis was −0.93–0.18, which demonstrates the dataset’s multivariate normal distribution. Table 4 reveals the goodness of fit of items. The indices χ^2^ (1.90), RMSEA (0.09), TLI (0.82), and CFI (0.82) met the goodness-of-fit criteria, whereas the other indices did not [23,24]. Factor loadings of each item and error ratios are presented Figure 2.

The scale’s convergent validity was confirmed using CR and AVE (Table 5). The CR of the SSNER-SL was 0.93–0.88, whereas the AVE was 0.56–0.64; these satisfy the requirement of a CR higher than 0.70 and AVE higher than 0.80 [25].

Table 6 depicts the results of discriminant validity testing. The square root of AVE ranged from 0.75 to 0.80, and correlation coefficients ranged from 0.63 to 0.76. These values confirmed the scale’s discriminant validity (Table 6).

#### 3.4.5. Reliability Testing

The Cronbach’s alpha values for the four factors were 0.96 for clinical mentorship, 0.93 for research, 0.88 for advising, and 0.88 for teaching, and the value was 0.97 for the overall SSNER-SL of the 39-item final scale (Table 5). Thesplit-half reliabilities were 0.94 and 0.96, respectively. The ITC was from 0.42 to 0.78, and Cronbach’s alpha if item deleted was 0.97 with final items. The Cronbach’s alpha values for item deletion in each of the factors ranged from 0.94 to 0.96.

### 3.5. Confirmation of the Final Instrument

After validation testing, a total of 39 items and four factors were identified as follows: clinical mentorship (18 items), research (10 items), advising (5 items), and teaching (6 items). The total score of nursing educators’ role among Sri Lankan nursing educators was 3.20 ± 0.61 points. The most commonly performed nursing educators’ role was clinical mentorship (3.41 ± 0.71); it was followed by advising (3.35 ± 0.73) and teaching (3.24 ± 0.66). The least performed nursing educators’ role was research (2.79 ± 0.81) (Table 7).

## 4. Discussion

In this study, the SSNER-SL scale that reflects the Sri Lanka nursing context was developed. The development of the SSNER-SL scale involved six steps in accordance with the process proposed by DeVellis [11]. The scale development process described by DeVellis provides a high degree of confidence in the scale’s content and construct.

The initial step of scale development was customized using the roles of nursing educators discussed in the literature and local nursing experts’ opinions on nursing educators’ roles, which were collected through FGIs. For this purpose, several nursing educators in Sri Lanka were interviewed. Further, the educators evaluated the scale’s content validity, which was found acceptable. Facial validity evaluation determined the suitability of the items’ contents to the nursing education context of Sri Lanka [26,27]. Additionally, regarding content validity testing, the experts were asked for ideas on adequacy and alignments to increase the scale’s validity [11]. The contents of the role internationally required for nursing educators were reviewed by South Korean nursing experts who are working with the ODA project and have understanding of the Sri Lankan nursing environment. Therefore, it is accepted that the nursing educators’ role outlined in the SSNER-SL included content that could satisfy both the Sri Lankan and international contexts.

DeVellis [11] recommended a thorough explanation of a single concept can be provided by administering a tool with 37–40 items. The SSNER-SL included 39 items with an explanatory variance of 63.58%. Therefore, the SSNER-SL can be considered an instrument that efficiently measures the self-efficacy of nursing educators’ role in Sri Lanka.

Regarding construct validity, factor analysis is one of the strongest and most commonly used approaches to validate an instrument [28]. In this study, the EFA with a four-factor model and the CFA revealed a conceptually meaningful pattern of item loadings. DeVellis [11] suggested that the scale’s items are statements regarding the specific concept to be measured; therefore, it is ideal to match the relevant theory with an instrument. In this study, concept analysis identified four attributes of nursing educators’ roles: teaching, research, clinical mentorship, and advising, which were incorporated in the SSNER-SL. It followed the recommendation of DeVellis [11], consistent with the four attributes of the role of nursing educators.

Clinical mentorship, which was the first factor of SSNER-SL, was found to be the most commonly performed role. In Sri Lanka, nursing educators act as administrators and nurse leaders [5]. This study disclosed that 92.0% of the participants had a clinical experience of more than 5 years as nurses in major state hospitals. Even after the Baccalaureate degree in nursing started in 2018, most of the nursing schools remained under the control of the country’s Ministry of Health. Since the basic nursing schools record more than 7000 h per year of clinical practicum in Sri Lanka, the role of clinical mentorship remains very high [29]. Therefore, clinical mentorship is a very familiar and easily adjustable role for nursing educators even at the novice level. It is extensively applied in nursing education worldwide, as well [30]. Nursing students should be trained to integrate their knowledge into practice [31]. Further, Chen et al. [30] identified professional development, psychosocial support, and learning facilitation as the three dimensions of the mentor role for integration. Therefore, the derivation of clinical mentoring as the scale’s first factor can be considered a reflection of the Sri Lankan nursing context.

Research, which was the second factor of the SSNER-SL, was found to be the least commonly performed role. Educators should conduct research by keeping themselves informed of the latest developments in their academic subjects and in the discipline of education [32,33]. Further, inquiry-oriented practices, such as having the capacity, motivation, and opportunity to use research-related skills, may help them enhance their professional practice as educationists [34]. According to Jayasekara and Amarasekara [3], two-thirds of the Sri Lankan nursing curriculum focuses on clinical practicum and emphasizes technical skills in hospital-based education, compared to the theory-based education provided by universities. Further, the current study revealed that 28.6% of the surveyed nursing educators were not academically qualified to assume nursing educators’ roles. Similarly, Chapman et al. [35] stated that most of the nursing educators in Vietnam were inadequately prepared for research. However, to date, very little attention has been paid to the research component of nursing education in Sri Lanka, and it is necessary to incorporate research in the country’s nursing education and training contexts [3]. The incorporation of research as the second factor is meaningful because the SSNER-SL clarifies the ideal roles of nursing educators.

The third factor of the SSNER-SL was advising. The nursing educators’ advising role significantly influences nursing students’ retention and success [36]. Further, from the students’ perspective, advising helps optimize the students’ occupational and personnel career development [37]. To provide effective academic advice to students, faculty should foster relationships based on trust with the students and use various online methods such as the simple notification service in conformity with the students’ needs [37].

Teaching was the last factor of the SSNER-SL. In recent years, the philosophy of learning has been changing from traditional teacher-centered learning to student-centered learning [38]. Since self-directed learning is becoming increasingly popular in developed countries, nursing educators in developing countries should be prepared to guide their students’ learning methods appropriately [29]. Accordingly, the role of nursing educators is rapidly changing from that of a knowledge transmitter to the role of a facilitator, knowledge navigator, or co-learner [39]. Hence, they should adopt various teaching strategies, styles, and evaluation methods to become good teachers [34]. To evaluate the effectiveness of teaching, a scale that can measure the teaching level of nursing educators should be developed.

The Cronbach’s alpha coefficients of the SSNER-SL and subfactors satisfied the criterion established by Nunnally and Bernstein [40] that the Cronbach’s alpha in instrument development should be greater than 0.70. In the case of a tool used in intervention research, the Cronbach’s alpha coefficient of 0.95 or higher is considered as a desirable standard instrument [41]. The Cronbach’s alpha coefficient of 0.97 in this study is meaningful because it is expected that the SSNER-SL will be used in further study to identify the effectiveness of interventions for improving nursing education system in developing countries, including Sri Lanka. On the other hand, in the case of the Cronbach’s alpha coefficient of 1.0, each item in the scale is considered to measure the same [42]. Therefore, it is necessary to reconfirm the reliability of SSNER-SL by repeating the test.

This study has a few limitations. The goodness of fit in CFA satisfied the criteria for standardized chi-square, RMSEA, TLI, and CFI measures but not the criteria for RMR, GFI, and NFI measures. The following are considered the reasons for this result: First, although 150 participants were considered eligible to participate, only 126 participants were included in this study. Second, to ensure the accuracy of CFA analysis, independent collection of data that do not overlap with EFA data is required [43,44]. However, the same data were analyzed for both CFA and EFA in this study. The reason for these limitations was the limited number of nursing educators in Sri Lanka that could satisfy the sample size required for this study. Additionally, the study did not test criterion-related validity. Because there are no data on other relevant scales, it is difficult to determine to what extent the SSNER-SL reflects the international nursing educator’s role. Moreover, the SSNER-SL should be retested to ensure test-retest reliability.

Despite these limitations, this study has the following strengths: it attempted to develop a scale to assess the roles of nursing educators in a developing country where the system of nursing education is undergoing a transition from the provision of diploma-level courses to that of a four-year undergraduate nursing program. Additionally, the SSNER-SL is a reliable tool since it has a construct validity with an explanatory variance more than 60% and a reliability with a Cronbach’s alpha coefficient more than 0.80.

## 5. Conclusions

This study developed and validated the SSNER-SL. This instrument was developed systematically according to steps outlined by DeVellis’s Scale Development Model. The validity and reliability of the four-factor, 39-item SSNER-SL were clearly established.

This instrument will be useful in all research that measures the self-efficacy of nursing educator’s roles in developing countries with similar context to that of Sri Lanka. Furthermore, the content of each item in SSNER-SL may be used as a valuable guideline to specify nursing educators’ roles in developing countries. SSNER-SL is a tool to measure the overall efficacy of nursing education. Based on this study, it is proposed to develop scales for clinical practicum education applicable to developing countries similar to Sri Lanka.

## Figures and Tables

**Figure 1 ijerph-18-07773-f001:**
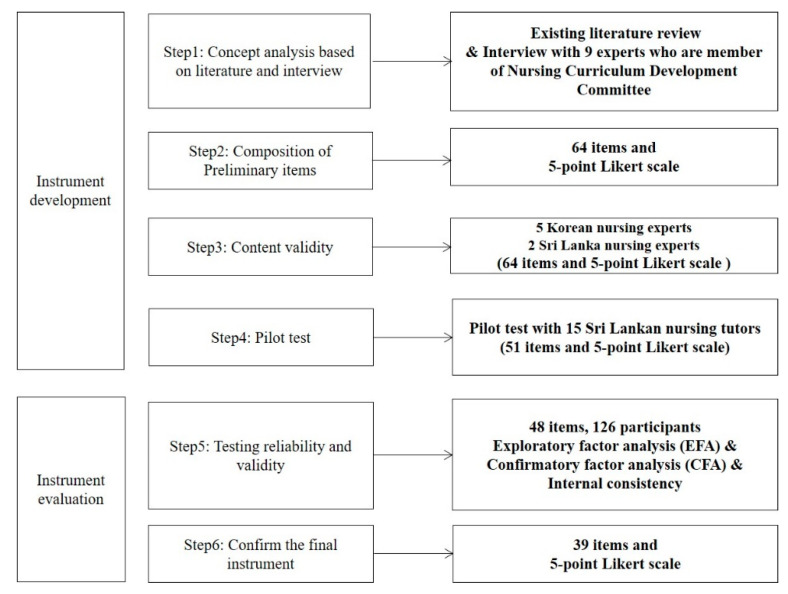
Scale development process.

**Figure 2 ijerph-18-07773-f002:**
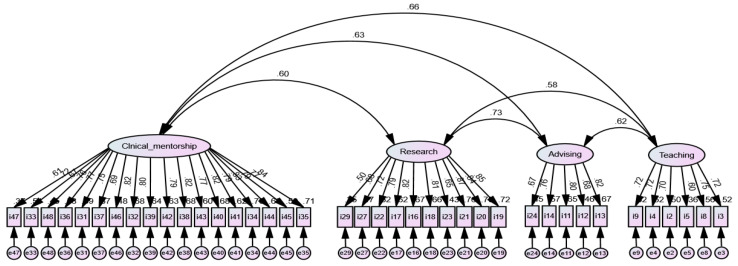
Confirmatory factor analysis for SSNER-SL.

**Table 1 ijerph-18-07773-t001:** Four themes extracted from concept analysis.

Theme	Category	Contents
Teaching	Student assessment	⬥ Development of an objective assessment tool ⬥ Development of a practical examination
	Curriculum development	⬥ Designing course work ⬥ Delivering curriculum contents
	Student support	⬥ Motivating students ⬥ Guiding students to overcome problems
	Classroom lecture	⬥ Planning teaching methodologies ⬥ Guiding students to perform self-learning
Research	Engagement in research activities	⬥ Creating a national research problem ⬥ Publishing research results
	Generation of new knowledge	⬥ Dissemination of research findings ⬥ Utilization of current literature
	Perform life-long learning	⬥ Nurturing intrinsic motivation
Clinical mentorship	Practice in the real word	⬥ Delivering care ⬥ Facing emerging challenges in clinic
	Training nursing skills	⬥ Preparing students ⬥ Demonstrating nursing skills ⬥ Evaluating nursing students
	Applying new knowledge in clinical practice	⬥ Facilitating evidence-based practice
Advising	Relationship with other members	⬥ Maintaining relationships with members
	Communication	⬥ Applying communication skills
	Developing leadership	⬥ Remaining alert to latest global trends and issues

**Table 2 ijerph-18-07773-t002:** General characteristics of participants (*n* = 126).

Characteristics		*n*	%	Mean ± SD	Min–Max.
Sex	Female	102	81.0		
	Male	24	19.0		
Age (year)				46.36 ± 5.21	37–59
Institution	Basic nursing school	117	92.9		
	PBCN	9	7.1		
Maximum qualification	Post-graduation	19	15.1		
	Bachelor’s degree	71	56.3		
	Diploma	36	28.6		
Teaching experience (year)				10.50 ± 3.16	4–18
Clinical nursing experience (year)				9.87 ± 4.52	4–36
Types of clinical settings ^1^	National hospital	22	17.5		
	Teaching hospital	86	48.9		
	General hospital	43	24.4		
	District general hospital	25	14.2		

^1^ Multiple response, PBCN = post basic college of nursing, SD = standard deviation.

**Table 3 ijerph-18-07773-t003:** Factor loadings of the second exploratory factor analysis (*n* = 126).

Items ^1^		Communality	Factor 1	Factor 2	Factor 3	Factor 4
	I Have the Ability to …		Clinical Mentorship	Research	Advising	Teaching
35	Build relationships based on mutual trust and respect with clinical staff.	0.74	**0.80**	0.12	0.25	0.14
45	Provide constructive feedback to students on their clinical performance.	0.42	**0.80**	0.06	0.07	0.16
44	Monitor the improvement of studies, clinical learning, and activities.	0.68	**0.80**	0.15	0.09	0.13
34	Encourage students to adapt themselves to their clinical nursing career.	0.74	**0.77**	0.13	0.32	0.18
41	Utilize appropriate clinical learning opportunities according to students’ stage of learning.	0.66	**0.77**	0.18	0.14	0.12
40	Provide support to students to facilitate their transition from one clinical learning environment to another.	0.69	**0.76**	0.18	0.21	0.19
43	Encourage students to combine theory and practice during clinical placement.	0.62	**0.74**	0.10	0.26	0.15
38	Contribute to the organization of a learner-friendly clinical environment.	0.70	**0.73**	0.31	0.07	0.27
42	Guide students to reflect on their clinical experiences.	0.64	**0.72**	0.24	0.21	0.14
39	Orient students to understand the legal framework of patient care management.	0.66	**0.72**	0.19	0.16	0.28
32	Maintain new knowledge of clinical practice.	0.69	**0.72**	0.33	0.20	0.20
46	Evaluate students’ clinical learning by using various clinical assessment strategies.	0.61	**0.70**	0.26	−0.12	0.18
37	Prioritize patient-centered care to support students’ clinical learning.	0.61	**0.69**	0.18	0.33	0.09
31	Cooperate in academic management functions.	0.64	**0.68**	0.31	0.09	0.26
36	Create opportunities for students to build relationships with clinical staff.	0.60	**0.68**	0.27	0.25	0.10
48	Be attentive to the emotional stress faced by students in clinical situations.	0.50	**0.66**	0.05	0.24	0.15
33	Develop professional relationships with the members of other academic communities.	0.57	**0.64**	0.37	0.03	0.17
47	Direct students to search the literature that is relevant to their nursing interventions.	0.42	**0.58**	0.18	0.09	0.21
19	Engage in research activities in one’s interest areas.	0.77	0.11	**0.86**	0.17	0.12
20	Create rational research problems/arguments.	0.76	0.15	**0.85**	0.13	0.14
21	Utilize evidence based on knowledge or research findings.	0.78	0.18	**0.83**	0.24	0.08
23	Build relationships with national and international professional organizations.	0.53	0.14	**0.71**	0.08	0.11
18	Remain alert to latest global trends and issues.	0.65	0.34	**0.69**	0.21	0.19
16	Perform life-long learning to increase professional competency as a nurse educator.	0.71	0.30	**0.66**	0.42	0.10
17	Utilize relevant and current literature in all academic activities.	0.66	0.24	**0.63**	0.40	0.21
22	Safely, responsibly, and ethically handle the information obtained over the Internet.	0.60	0.27	**0.58**	0.44	0.05
27	Analyze policy guidelines or circulars relevant to nursing education.	0.57	0.30	**0.58**	0.24	0.28
29	Participate in interdisciplinary actions to address healthcare and educational issues and needs.	0.40	0.31	**0.50**	0.02	0.22
13	Act as a role model for students’ professional socialization.	0.65	0.27	0.25	**0.75**	0.15
12	Use linguistic skills (English) in reading, writing, listening, and speaking for academic purposes.	0.62	0.13	0.19	**0.73**	0.18
11	Provide effective counseling and help to students to solve their needs and problems.	0.66	0.24	0.26	**0.72**	0.22
14	Use feedback gained from oneself, peers, students, and administrators to improve role effectiveness.	0.55	0.26	0.35	**0.59**	0.20
24	Be accountable for one’s own professional judgments, actions, and outcomes.	0.59	0.25	0.43	**0.55**	0.11
3	Implement various teaching strategies, such as active learning, problem-based learning, and team-based learning, in the classroom.	0.70	0.21	0.28	−0.06	**0.78**
8	Design objective assessment tools, such as checklists, test papers, assignments, portfolios, and clinical diaries.	0.60	0.23	0.17	0.24	**0.71**
5	Guide students to perform self- and peer evaluations of academic activities.	0.54	0.15	0.00	0.26	**0.67**
2	Plan teaching methodologies suited to students’ learning needs and styles.	0.60	0.35	0.19	0.01	**0.67**
4	Motivate students to use various learning sources, such as print materials, audio visual materials, the Internet, and smartphones.	0.59	0.26	0.25	0.26	**0.61**
9	Use students’ assessment and evaluation data to improve the teaching–learning process.	0.52	0.32	0.17	0.29	**0.59**
	Explained variance		10.65	6.45	3.95	3.74
	Explained percentage		27.31	16.55	10.12	9.60
	Cumulative percentage		27.31	43.86	53.98	63.58

^1^ The analysis was performed on 39 items.

**Table 4 ijerph-18-07773-t004:** Model fit of the SSNER-SL (*n* = 126).

Goodness of Fit	χ^2^(*p)*	χ^2^/df	RMR	RMSEA	GFI	NFI	TLI	CFI
Baseline	*p* > 0.05	≤3	≤0.05	≤0.10	≥0.90	≥0.80	≥0.80	≥0.80
Results	1321.2 (*p* < 0.001)	1.90	0.07	0.09	0.66	0.71	0.82	0.84

RMR = root mean square residual, RMSEA = root mean square error of approximation, GFI = goodness-of-fit index, NFI = normed fit index, TLI = Tucker–Lewis index, CFI = comparative fit index.

**Table 5 ijerph-18-07773-t005:** Confirmatory factor analysis results of the SSNER-SL (*n* = 126).

Factor	Item	Standardized Factor Loading	Critical Ratio	*p*	CR	AVE	Cronbach’s Alpha
Clinical mentorship	31	0.77			0.97	0.64	0.96
	32	0.83	10.20	<0.001			
	33	0.72	8.57	<0.001			
	34	0.83	10.34	<0.001			
	35	0.84	10.43	<0.001			
	36	0.76	9.23	<0.001			
	37	0.75	9.12	<0.001			
	38	0.82	10.16	<0.001			
	39	0.80	9.86	<0.001			
	40	0.82	10.16	<0.001			
	41	0.79	9.59	<0.001			
	42	0.79	9.71	<0.001			
	43	0.77	9.40	<0.001			
	44	0.79	9.70	<0.001			
	45	0.77	9.34	<0.001			
	46	0.69	8.21	<0.001			
	47	0.61	7.14	<0.001			
	48	0.67	7.95	<0.001			
Research	16	0.82			0.93	0.56	0.88
	17	0.79	10.34	<0.001			
	18	0.81	10.75	<0.001			
	19	0.85	11.55	<0.001			
	20	0.84	11.26	<0.001			
	21	0.87	11.98	<0.001			
	22	0.72	9.14	<0.001			
	23	0.65	8.01	<0.001			
	27	0.68	8.48	<0.001			
	29	0.50	5.86	<0.001			
Advising	11	0.80			0.88	0.61	0.88
	12	0.68	7.84	<0.001			
	13	0.82	9.92	<0.001			
	14	0.76	9.02	<0.001			
	24	0.67	7.80	<0.001			
Teaching	2	0.70			0.89	0.57	0.97
	3	0.72	7.354	<0.001			
	4	0.72	7.345	<0.001			
	5	0.60	6.186	<0.001			
	8	0.75	7.612	<0.001			
	9	0.72	7.363	<0.001			

AVE = average variance extracted; CR = construct reliability.

**Table 6 ijerph-18-07773-t006:** Correlation matrix and the square root of average variance extracted for discriminant validity of SSNER-SL (*n* = 126).

Sub-Factors of the SSNER-SL	Factor 1	Factor 2	Factor 3	Factor 4
Factor 1: Clinical mentorship	0.80 ^1^			
Factor 2: Research	0.63 (<0.001)	0.75 ^1^		
Factor 3: Advising	0.76 (<0.001)	0.71 (<0.001)	0.78 ^1^	
Factor 4: Teaching	0.69 (<0.001)	0.73 (<0.001)	0.67 (<0.001)	0.75 ^1^

^1^ Square root average variance extracted, SSNER-SL, Self-Efficacy Scale for Nursing Educator’s Role in Sri Lanka.

**Table 7 ijerph-18-07773-t007:** Self-efficacy of nursing educators’ roles in Sri Lanka (*n* = 126).

Factor (Number of Items)	Mean ± SD	Min-Max	Rank
SSNER-SL (39)	3.20 ± 0.61	2.10–4.83	—
- Clinical mentorship (18)	3.41 ± 0.71	1.89–5.0	1
- Research (10)	2.79 ± 0.81	1.40–4.90	4
- Advising (5)	3.35 ± 0.73	1.60–5.0	2
- Teaching (6)	3.24 ± 0.66	1.83–4.67	3

SSNER-SL = Self-Efficacy Scale for Nursing Educator’s Role in Sri Lanka, SD = standard deviation.

## Data Availability

The data presented in this study are available on request from the corresponding author. The data are not publicly available due to privacy concerns.

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
