# Peer review of "Development and Validation of a Self-Efficacy Scale for Nursing Educators’ Role in Sri Lanka"

_ijerph, 2021, doi:10.3390/ijerph18157773_

Round 1

Reviewer 1 Report

 Please enter your comments and suggestions for authors:

Consider that the summary could be less extensive, and give greater importance to the consequences that this study provides for the training and evaluation of the clinical practices of nursing students.

Reviewer 2 Report

The present study developed a 39-item scale to assess level of self-efficacy among Sri Lankan nursing tutors.

There are two main concerns regarding the evaluation of validity and reliability of the scale.

  1. Validity: there is no information on criteria-related validity of the scale, e.g. confirmation of expected differences in level of self-efficacy according to respondents' characteristics, or expected correlations with preexisting scales. Absence of data with other relevant scales also makes it difficult to determine to what extent the developed scale reflects country-specific nursing competency.
  2. Reliability: test-retest reliability is not evaluated. 

Reviewer 3 Report

Introduction

Lines 50-53: you report that nursing tutors are not prepared to monitor nursing students. Before implementing a self-efficacy instrument on nursing tutors, was a program of clinical supervision training for teachers/educators and tutors performed, in order to explain what would be expected of them, when tutoring students?

I believe a definition should be provided on the nursing educator and the nursing tutor concepts. Are they synonyms?

Methods

Lines: 102-102: You refer to interviews as being recorded but were they subjected to content analysis? Can you describe the process?

Lines 106-108: it no clear how data from the literature review and from interviews were grouped to formulate preliminary instrument questions.

Lines 110-114: why were experts from Korea in larger number than Sri Lanka nursing experts?

Line 122: the Likert scale was 4-point or 5-point?

Sample and testing: this instrument was built for Sri Lankan’s tutors. Could you explain why were the participants’ unfamiliar with the Sri Lankan research culture?

Data analysis

EFA should include analysis of the scree plot or a parallel analysis to allow a proper understanding of the option for 4 dimensions. Eigenvalue analysis is an insufficient approach to this analysis. Anti-image matrix values should also be included in the analysis. Please refer to the minimum value, as this could help in understanding the fitness of the items for EFA. Apart from these requirements, the analysis is well built and interesting.

Regarding reliability analysis, a score of 0.97 is problematic as it may reveal that all the items are analyzing the same dimension. This should be addressed through the eyes of the researcher, particularly discussing if this should be addressed through continuous testing of the scale in question, and/or how are these items different to justify the inclusion in the final version. Please also refer to how Cronbach’s alpha varies due to item deletion in each of the factors.

CFA could benefit from the inclusion (or reference) to multivariate outliers (and consequent removal) through analysis of the squared Mahalanobis distance. The inclusion of the multivariate normality would increase the quality of the analysis. There is no reference to a potential analysis of the modification indices that could help increase the model’s quality through the inclusion of covariances between errors. If possible, including the image resulting from AMOS would help the reader. Otherwise, very good analysis of the CFA results.

Conclusion

After developing and implementing a self-efficacy instrument, what is the main goal? What will you do with these results? The scale identifies fragility, but in what way do you consider to have the potential to improve tutors self-efficacy?

Above all, and seeing this is an international journal, how can your specific culture results help in a global perspective. This will increase scientific soundness and interest to the readers.

Round 2

Reviewer 2 Report

No further additions to previous comments and suggestions.
